# Molecular Mechanisms of Ferroptosis and Updates of Ferroptosis Studies in Cancers and Leukemia

**DOI:** 10.3390/cells12081128

**Published:** 2023-04-11

**Authors:** Hiroki Akiyama, Bing Z. Carter, Michael Andreeff, Jo Ishizawa

**Affiliations:** Section of Molecular Hematology and Therapy, Department of Leukemia, The University of Texas MD Anderson Cancer Center, Houston, TX 77030, USA; hakiyama1@mdanderson.org (H.A.); bicarter@mdanderson.org (B.Z.C.); mandreef@mdanderson.org (M.A.)

**Keywords:** ferroptosis, leukemia, iron, phospholipid, lipid peroxidation, p53

## Abstract

Ferroptosis is a mode of cell death regulated by iron-dependent lipid peroxidation. Growing evidence suggests ferroptosis induction as a novel anti-cancer modality that could potentially overcome therapy resistance in cancers. The molecular mechanisms involved in the regulation of ferroptosis are complex and highly dependent on context. Therefore, a comprehensive understanding of its execution and protection machinery in each tumor type is necessary for the implementation of this unique cell death mode to target individual cancers. Since most of the current evidence for ferroptosis regulation mechanisms is based on solid cancer studies, the knowledge of ferroptosis with regard to leukemia is largely lacking. In this review, we summarize the current understanding of ferroptosis-regulating mechanisms with respect to the metabolism of phospholipids and iron as well as major anti-oxidative pathways that protect cells from ferroptosis. We also highlight the diverse impact of p53, a master regulator of cell death and cellular metabolic processes, on the regulation of ferroptosis. Lastly, we discuss recent ferroptosis studies in leukemia and provide a future perspective for the development of promising anti-leukemia therapies implementing ferroptosis induction.

## 1. Introduction

Regulated cell death (RCD) is essential for ontogenetic development and homeostasis, and its dysregulation can result in various pathological conditions, including cancer [1,2]. Apoptosis is the best-established mode of RCD and was first characterized in 1972 by Kerr et al. in both physiological and pathological contexts [3]. Because most cancer cells depend on the dysregulation of apoptosis pathways for their survival, therapeutic interventions targeting these pathways have been studied extensively [4]. During the past few decades, however, numerous studies have uncovered additional modes of RCD whose morphologic features and mechanisms are distinct from those of apoptosis [1,2,5].

Ferroptosis is one of the non-apoptotic modes of RCD characterized by a dependency on iron and lipid peroxidation. The term “ferroptosis” was first coined in 2012 by Dixon et al. as an iron-dependent form of non-apoptotic cell death induced by the RAS-selective lethal small molecules erastin and (1S,3R)-RSL3 (RSL3) [6]. Mechanistically, erastin blocks the uptake of cystine through system x_c_^−^ and depletes cellular glutathione (GSH). Two years later, Yang et al. discovered that glutathione peroxidase 4 (GPX4) is a target of RSL3 and an essential regulator of ferroptosis in many cancer cell types [7]. Since then, ferroptosis has drawn extensive attention in a variety of biomedical research fields, including anti-viral or anti-tumor immune responses, ischemic organ injury, as well as cancer (detailed review elsewhere [8]). Ferroptosis has also been demonstrated to play important physiological roles, such as erythroid maturation [9], platelet production [10], aging [11], and immune cell functions [12,13,14]. The number of publications related to ferroptosis has grown exponentially over the last decade [5,15]. In particular, ferroptosis induction could be a novel therapeutic strategy for cancers that are resistant to apoptosis and conventional chemotherapy [16,17,18,19]. However, the regulatory pathways and cell death execution machinery of ferroptosis are not yet fully understood. In fact, recent studies indicate a highly variable and context-dependent nature of ferroptosis regulation in cancers [20,21].

Leukemia is characterized by the excessive proliferation, increased survival, and dysregulated differentiation of immature cells originating from hematopoietic stem and progenitor cells. With dysregulation of apoptosis mechanisms being essential components of leukemia development, progression, and therapy resistance, discovery of the BCL-2 inhibitor venetoclax transformed the therapeutic landscape of acute myeloid leukemia (AML) as well as that of chronic lymphocytic leukemia (CLL) [22,23,24]. Nevertheless, a substantial number of patients exhibit primary or acquired resistance to venetoclax-based therapies indicating cell-intrinsic and/or therapy-induced mechanisms of apoptosis resistance, particularly in AML cells [25]; therefore, alternative therapeutic strategies are urgently needed.

In this review, we summarize the current understanding of the regulatory mechanisms of ferroptosis, most of which have been identified through solid cancer studies, and its potential as a novel concept for the therapy of leukemia.

## 2. Molecular Mechanisms of Ferroptosis Induction

The health and survival of most living organisms depend on the tight regulation of cellular reactive oxygen species (ROS), which are highly reactive byproducts of metabolic processes involving oxygen [26]. Excessive ROS can affect various cellular components, including membrane phospholipids, leading to lipid peroxidation and membrane damage [27]. An accumulation of lipid peroxides overwhelms the cells’ antioxidative defense mechanisms, leading to ferroptosis [28,29]. Major pathways and molecular players in the regulation of ferroptosis are summarized in Table 1 and discussed in detail in this section and the following Section 3.

### 2.1. Ferroptosis Regulation through Phospholipid Metabolism

Phospholipids containing polyunsaturated fatty acids (PUFAs) are the major substrates for lipid peroxidation and are far more susceptible to lipid peroxidation than saturated fatty acids (SAs) or mono-unsaturated fatty acids (MUFAs). Therefore, phospholipid metabolism plays a critical role in ferroptosis regulation (Figure 1). Hydroxyl radicals or lipoxygenases attack the weak C-H bonds between the adjacent C=C double bonds of PUFAs to remove a hydrogen atom and initiate lipid peroxidation reactions [30].

PUFAs are primarily obtained from dietary intake and/or a series of elongation and desaturation reactions [31]. Mammalian cells do not have a de novo system of synthesizing major PUFAs (n-3 and n-6 PUFAs) and thus require linolenic acid (C18:3 [n-3]) or linoleic acid (C18:2 [n-6]) as dietary precursors for the subsequent synthesis of other PUFAs. Enzymes required for PUFA synthesis include fatty acid desaturases (e.g., FADS1 and FADS2) and fatty acid elongases (e.g., ELOVL5), which have been identified as ferroptosis enhancers [32,33]. These PUFAs were shown to preferentially undergo lipid peroxidation to induce ferroptosis in an acidic cancer cell environment [34]. In contrast, the energy sensor adenosine monophosphate-activated protein kinase (AMPK) works as a ferroptosis suppressor by negatively regulating acetyl-CoA carboxylase (ACC) and following PUFA elongation [35]. Lipid droplets, which store fatty acids as triacylglycerols and sterol esters, can sequester PUFA to prevent lipid peroxidation [36]. Therefore, the activation of enzymes required for lipid droplets biogenesis or blocking autophagic degradation of lipid droplets (lipophagy) suppresses ferroptosis [37]. This study indicates the mechanistic link between lipid homeostasis regulated through lipid droplet metabolism and ferroptosis.

Extensive evidence shows that PUFAs must be incorporated into membrane phospholipids for ferroptosis to be induced through peroxidation. In particular, acyl-CoA synthase 4 (ACSL4) preferentially recognizes arachidonic acid (C20:4 [n-6]) as a substrate to mediate the esterification of fatty acyl-CoA and thus promotes the incorporation of PUFAs into phospholipids. Consistently, ACSL4 was identified as an essential gene for GPX4 inhibition–induced ferroptosis through CRISPR-based screens [38,39]. Another genetic screening study showed that lysophosphatidylcholine acyltransferase 3 (LPCAT3), an enzyme that transfers fatty acyl-CoA into phospholipids, is also required for ferroptosis [40]. However, in a recent study comparing 24 large-scale genetic screens, Magtanong et al. discovered that essentialities of previously reported ferroptosis regulators are highly context-specific [21]. Indeed, the disruption of ACSL4 or LPCAT3 suppresses GPX4 inhibition–induced ferroptosis, but not erastin2-induced ferroptosis, which suggests distinct mechanisms of ferroptosis execution. In contrast, MUFA-containing phospholipids suppress membrane lipid peroxidation and ferroptosis. MUFAs are synthesized from SAs by stearoyl-CoA desaturase 1 (SCD1) and then incorporated into phospholipids by ACSL3. In fact, the inhibition of SCD1 could induce ferroptosis through a decrease in membrane MUFAs and a compensatory increase in PUFAs [41]. In contrast, the exogenous supplementation of MUFAs protects cells from ferroptosis in an ACSL3-dependent manner [42].

In addition, enzymes in peroxisomes, which are involved in ether-linked phospholipids (termed “plasmalogens”) could also play an important role in ferroptosis. Plasmalogens contain PUFA and are prone to lipid peroxidation; thus enzymes required for plasmalogen synthesis (alkylglycerone phosphate synthase [AGPS], fatty acyl-CoA reductases 1 [FAR1], and 1-acylglycerol-3-phosphate O-acyltransferase 3 [AGPAT3]), as well as for peroxisome biogenesis (peroxisomal biogenesis factor 10 [PEX10] and PEX3), reportedly promote ferroptosis [43,44]. Of note, peroxisomes are known to exert diverse functions depending on organisms and cell types, whose role in leukemia cell lipid metabolism remains largely unknown.

### 2.2. Iron Metabolism and Ferroptosis

#### 2.2.1. Iron Homeostasis and the Regulation of Labile Iron Pool

Iron is an essential element in cellular processes, and its balance is maintained by the tight regulation of its cellular uptake, recycling, and export [45]. Cytoplasmic iron forms a labile iron pool (LIP), a dynamic pool of redox-active iron that is essential for lipid peroxidation and ferroptosis (Figure 2).

First, ferric ion (Fe^3+^) bound to transferrin (TF) in the plasma is taken up by cells through transferrin receptor 1 (TFR1)–mediated endocytosis [46]. Independent of the canonical iron uptake pathway through TF-TFR1 interaction, a recent study demonstrated that CD44 uptakes hyaluronates-bound iron in multiple types of solid cancer cells, especially those with epithelial-mesenchymal transition, to fulfill higher iron needs in these cells [47]. Once taken up, Fe^3+^ is released from the complex, reduced to ferrous iron (Fe^2+^), and transported to the cytoplasm by the divalent metal transporter 1 (DMT1) [48]. Fe^2+^ released in the cytoplasm forms LIP and is used for metabolic processes in each cellular compartment [49]. While LIP exists in association with various ligands, including GSH, to minimize its cytotoxicity [50], excess iron is either stored in ferritin or exported into circulation by ferroportin 1 (FPN1, also known as solute carrier family 40 member 1 [SLC40A1]), the only known mammalian iron exporter [51]. When the iron level is low, cytosolic ferritin is utilized as a source of cellular iron for various metabolic processes. The autophagic degradation of ferritin, “ferritinophagy,” causes intracellular iron levels to increase. Upon the activation of autophagic processes, the cargo receptor nuclear receptor coactivator-4 (NCOA4) binds ferritin and promotes lysosomal transportation and degradation, leading to the release of Fe^2+^ to increase the levels of LIP [52].

Mitochondria are the primary organelles that require iron for multiple metabolic processes [53,54]. Cellular iron, in the form of either Fe^2+^ (in LIP) or Fe^3+^ (in Fe^3+^-TF complex), is transported into the mitochondrial matrix by several mechanisms involving DMT1 [55] at the outer membrane and mitoferrins (MFRNs) at the inner membrane [56]. Within the matrix, iron is primarily utilized for the biosynthesis of iron-sulfur complexes (ISCs) and heme. ISCs are essential for several proteins, including electron transport chain complexes addition, while the heme oxygenase 1 (HO-1)–mediated degradation of heme is important for the recycling and maintenance of cellular iron, including LIP [57].

#### 2.2.2. Essential Roles of LIP in Lipid Peroxidation

Iron plays crucial roles in ROS formation and lipid peroxidation both directly and indirectly [58]. Cytoplasmic Fe^2+^, or LIP, catalyzes non-enzymatic hydroxyl radical (OH˙) formation via the Fenton reaction (Fe^2+^ + H_2_O_2_ → Fe^3+^ + OH^−^ + OH˙). In addition, in the presence of superoxide radicals (O_2_˙^−^), which are mainly produced through mitochondrial respiration, Fe^3+^ undergoes the Harber--Weiss reaction (Fe^3+^ + O_2_˙^−^ → Fe^2+^ + O_2_) to further promote the Fenton reaction. Compared with hydrogen peroxide (H_2_O_2_), superoxide, and other ROS, the hydroxyl radical that these reactions produce is highly reactive with various biological molecules, including lipids, proteins, and DNAs. These radicals remove a hydrogen atom from the bisallylic methylene of a PUFA, leading to the formation of a lipid-peroxyl radical (PLOO˙) and a lipid peroxide (PLOOH) [59]. Alternatively, LIP can also work as a cofactor for many enzymes that produce ROS, including nicotinamide adenine dinucleotide phosphate (NADPH) oxidases (NOXs), cytochrome P450 oxidoreductases (POR), nitric oxide synthases, lipoxygenases, and components of the mitochondrial electron transport chain. In particular, arachidonate lipoxygenases (ALOXs) directly catalyze lipid peroxidation in a highly regulated manner [60]. Enzymatic lipid peroxidation controls oxygen insertion, resulting in the formation of specific enantiomers. In contrast, nonenzymatic lipid peroxidation progresses without stereochemical control, leading to the formation of racemic and regioisomeric lipid peroxides, which are distinguishable from the products of enzymatic lipid peroxidation. The differential roles of enzymatic and nonenzymatic lipid peroxidation, particularly in the execution of ferroptosis, remain to be fully elucidated [61].

#### 2.2.3. Ferroptosis Regulation through Iron Metabolism

The regulation of cellular iron uptake involves TFR1 and the Fe^3+^-TF complex and affects ferroptosis. TFR1 knockdown decreases cellular iron and suppresses ferroptosis, whereas TF expression is required for ferroptosis induction [62]. Interestingly, TFR1 has been identified as a specific marker of ferroptosis [63], but its sensitivity and specificity in various contexts of ferroptosis have yet to be validated. The knockdown of the iron exporter FPN1 promotes ferroptosis [64,65]. In contrast, the knockdown or overexpression of the iron transporter DMT1 suppresses or enhances ferroptosis, respectively [66,67]. Ferritin stores excess intracellular iron and helps protect cells from highly reactive LIP. The knockdown of ferritin heavy chain 1 (FTH1) facilitates iron overload–associated cardiomyopathy through ferroptosis [68]. Ferroptotic stimuli themselves have been shown to induce the expression of prominin2 (PROM2) to enhance the exosomal export of ferritin to protect cells from ferroptosis [69].

Ferritinophagy, which degrades ferritin to increase LIP and thereby promotes ferroptosis, is a mechanistic link between autophagy and ferroptosis [70]. The inhibition of ferritinophagy by the blockade of the autophagy pathway or the knockdown of NCOA4 suppresses ferroptosis [71,72]. Besides ferritinophagy, the HO-1–mediated degradation of heme is another source of cellular LIP and is related to ferroptosis. Erastin induces the expression of HO-1, which promotes ferroptosis in cancer cells [73]. Conversely, the inhibition of HO-1 protects cardiac cells from doxorubicin-induced ferroptosis [74]. However, the knockout of HO-1 has also been reported to enhance the ferroptosis of renal proximal tubular cells, which suggests that the role of HO-1 in ferroptosis depends on context [75].

Mitochondrial iron also plays an important role in the regulation of ferroptosis. The overexpression of FTMT has been reported to cause iron to shift from the cytosol to mitochondria, and to reduce LIP and protect cells from ferroptosis under various pathologic conditions [76,77,78,79]. A metabolism-focused CRISPR screening using Jurkat leukemia cells revealed that the loss of either of two mitoferrin isoforms, MFRN-1 (SLC25A37) or MFRN-2 (SLC25A28), sensitizes cells to GPX4 inhibition–induced ferroptosis [20], which suggests that the mitochondrial import of iron (perhaps because it is sequestered in the mitochondrial matrix) is essential to protecting cells from ferroptosis. However, the inhibition of GPX4 or system x_c_^−^ in normal liver stellate cells has been shown to promote mitochondrial p53 translocation and the activation of MFRN-2 to induce ferroptosis, which can be abrogated by the knockdown of MFRN-2 [80]. (The discrepancy between these two studies is another example of the context dependency of ferroptosis.) Like mitochondrial iron storage, ISC biosynthesis helps protect cells from ferroptosis by decreasing LIP levels. Frataxin (FXN) [81] and NFS1 cysteine desulfurase [82] are both required for ISC synthesis and are overexpressed in cancer cells. These proteins promote cancer cell growth and confer resistance to ferroptosis. Heme synthesis has also been reported to regulate doxorubicin-induced ferroptosis in cardiac cells. Doxorubicin induces mitochondrial iron accumulation by disrupting 5′-aminolevulinate synthase 1 (ALAS1), the rate-limiting enzyme in heme synthesis. The overexpression of ALAS1 protects cells from doxorubicin-induced ferroptosis [83].

Iron can also facilitate ferroptosis through activation of enzymes that catalyze lipid peroxidation. Among other iron-containing enzymes, arachidonic 15-lipoxygenase (ALOX15) was initially reported to be essential for ferroptosis [38,84]. In addition, the scaffolding protein phosphatidylethanolamine-binding protein 1 (PEBP1) regulates ALOX15 to enhance the peroxidation of phosphatidylethanolamine and induce ferroptosis in the setting of GPX4 inhibition [85]. However, in later studies, the knockout of ALOX15 failed to rescue GPX4 inhibition–induced ferroptosis in different cell models [86,87], indicating alternative pathways of lipid peroxidation. A recent CRISPR screen revealed that P450 oxidoreductase (POR) is involved in the enzymatic lipid peroxidation that leads to ferroptosis [88]. Arachidonic 12-lipoxygenase (ALOX12) has also been reported to be important for p53-dependent ferroptosis [89]. Therefore, the enzymatic regulation of lipid peroxidation and its involvement in ferroptosis execution depends on context and warrants further investigation.

### 2.3. What Is the Direct Cause of Ferroptosis?

Despite a decade of extensive study, the direct cause of ferroptotic cell death induced by iron-dependent lipid peroxidation has yet to be fully elucidated.

Lipidic aldehydes, which are generated as a result of or during PUFA peroxidation, are cytotoxic effectors of ferroptosis. Lipid hydroperoxides are the first PUFA metabolites that become unstable and degrade into reactive α, β-unsaturated aldehydes, such as 4-hydroxynonenal (4-HNE), which is derived from linoleic acids (C18:2 [n-6]) and arachidonic acids (C20:4 [n-6]), and 4-hydroxyhexenal (4-HHE), which is derived from docosohexaenoic, eicosapentaenoic, and linolenic acids (C22:6 [n-3], C20:5 [n-3], and C18:3 [n-3], respectively) [90]. Aldehyde toxicity is characterized by cell-localized micronutrient deficiencies in sulfur-containing antioxidants, including GSH, vitamins (B1, B6, folate), zinc, and retinoic acid, which cause oxidative stress and a cascade of metabolic disturbances. Aldehydes also react and form adducts with selective proteins in the cell, or can affect proteins in neighboring cells as exogenous aldehydes [91], causing protein loss-of-function and aggregation [92,93]. Reactive aldehydes also form adducts with DNA, inducing DNA strand breaks. Aldehyde dehydrogenases (ALDHs) detoxify lipidic aldehydes, including 4-HNE; thus, ALDH inhibitors may be useful as ferroptosis enhancers. Among the 19 ALDH isozymes, ALDH3A1 [94,95], ALDH3A2 [96], and ALDH2 [97] have been reported to exert anti-ferroptosis effects.

Recently, Pedrera et al. proposed that the formation of membrane nanopores, or “ferroptotic pores,” which is associated with cytosolic Ca^2+^ fluxes and precedes cytolysis upon ferroptotic stimuli, is a direct cause of cell death [98]. This Ca^2+^ flux induces the activation of the endosomal sorting complex required for transport (ESCRT)-III–dependent membrane repair machinery to counteract membrane rupture and ferroptotic cell death [99]. Ca^2+^ flux upon ferroptotic cell rupture was also suggested by Riegman et al., who demonstrated wave-like propagation of cell death between adjacent cells [100]. Conversely, cell-cell interactions play an important role also in the protection of cells against ferroptosis. For example, the activation of the E-cadherin–Hippo pathway modulates downstream Yes-associated protein 1 (YAP1) and the transcriptional coactivator with PDZ-binding motif (TAZ) signaling to suppress ferroptosis [101,102]. In addition to cytosolic Ca^2+^ transport, mitochondrial Ca^2+^ uptake via mitochondrial calcium uptake 1 (MICU1) has also been reported to be involved in cold stress–induced ferroptosis [103].

Further investigations of the molecular mechanisms directly involved in ferroptotic cell death as “a point of no return” are necessary to precisely detect and understand this unique cell death mode.

## 3. Mechanisms of Protection from Ferroptosis

### 3.1. The System x_c_^−^—GSH—GPX4 Axis

Ferroptosis was initially described as a form of cell death that is induced by erastin and RSL3 through the inhibition of cystine uptake via system x_c_^−^ and GPX4, respectively [6]. Given that cystine is an essential component of GSH, which is a co-factor of GPX4, the system x_c_^−^-GSH-GPX4 pathway was first established as the core defense mechanism against ferroptosis.

System x_c_^−^ is a heterodimeric plasma membrane cystine/glutamate antiporter composed of SLC7A11 (xCT) and SLC3A2 (CD98) [104]. System x_c_^−^ imports cystine while exporting intracellular glutamate. Imported cystine is reduced to cysteine, which is utilized for GSH biosynthesis. Cystine availability also promotes GPX4 protein synthesis through the activation of the mammalian target of rapamycin complex 1 (mTORC1) [105]. A minor portion of intracellular cysteine pool can also be synthesized de novo from methionine through the transsulfuration pathway [106]. The activation of this pathway through cystathionine β-synthase (CBS) [107] or S-adenosyl homocysteine hydrolase (SAHH) [108] confers resistance to ferroptosis. GSH is synthesized from cysteine via sequential enzymatic reactions involving the rate-limiting γ-glutamylcysteine synthetase (γ-GCS) and glutathione synthetase (GSS) [109]. The inactivation of system x_c_^−^ (e.g., by erastin) or the depletion of cystine from culture media has been reported to induce ferroptosis in a wide variety of cancer models [110,111,112]. Recently, Pardieu et al. studied the anti-leukemia effects of cystine depletion in AML and found that ferroptosis comprised only a part of the mechanisms of cell death induced [113]. The depletion of GSH also induces a mixture of cell death modes including ferroptosis [114,115]. Intriguingly, one recent study indicated that γ-GCS has a non-canonical, GSH-independent role in protecting cells from ferroptosis through the synthesis of γ-glutamyl peptides and the inhibition of glutamate stress [116]. In fact, glutamate metabolism is also known as a regulator of ferroptosis; the inhibition of SLC1A5-mediated glutamine uptake, the mitochondrial glutaminase (GLS2)–mediated synthesis of glutamate from glutamine, and the glutamic-oxaloacetic transaminase 1 (GOT1)–mediated synthesis of α-ketoglutarate from glutamate, can all suppress ferroptosis [62].

GPX4 is the only mammalian enzyme that catalyzes the reduction of lipid peroxides into nontoxic alcohols [117]. Since the discovery that RSL3 targets GPX4 to induce ferroptosis [7], a number of studies have established the roles of GPX4 as a central repressor of ferroptosis [105,118,119,120]. Importantly, therapy-resistant or -persistent solid cancer cells with a mesenchymal gene signature exhibit vulnerability to GPX4 inhibition, which suggests that targeting GPX4 has therapeutic potential in certain intractable phases or types of cancers [16,17]. Because GPX4 is a selenoprotein, its synthesis and activity are regulated by cellular selenium availability [121]. Therefore, the incorporation of selenium within GPX4 is also important for protecting cells from ferroptosis [122,123]. Mammalian GPX4 consists of three isoforms with distinct subcellular localizations: short, long, and nuclear isoforms of GPX4 (from here, we describe them simply as short, long, and nuclear GPX4). Whereas long GPX4 contains a mitochondria-targeting sequence and is transported into the mitochondrial matrix, short GPX4 mainly resides in the cytosol [124]. Of note, a portion of short GPX4 is translocated into the mitochondria in a mitochondria-targeted sequence–independent manner [125]. Only short GPX4 was initially believed to be important for cell protection against ferroptosis, mainly because short GPX4, but not the other GPX4 isoforms, rescues the lethal phenotype of GPX4 knockout in mice [125,126,127]. However, recent studies have demonstrated that long GPX4 also plays an important role in ferroptosis protection in doxorubicin-induced cardiomyopathy [128,129,130] and in cancer models under certain conditions, as described in the following Section 3.3 [131,132,133].

### 3.2. The Ferroptosis Suppressor Protein 1-Coenzyme Q Axis

In 2019, two groups independently reported a GPX4-independent mechanism of ferroptosis protection that involves ferroptosis suppressor protein 1 (FSP1, formerly known as apoptosis inducing factor mitochondria associated 2 [AIFM2]) [134,135]. FSP1 localizes on the plasma membrane and reduces ubiquinone, or coenzyme Q (CoQ), to ubiquinol (CoQH_2_). In fact, CoQ/CoQH_2_ works not only as an electron carrier in the mitochondrial electron transport chain complex but also as a major endogenous lipophilic radical-trapping antioxidant that protects cells from lipid peroxidation [136]. In this reaction, CoQH_2_ reduces lipid peroxides and is oxidized to CoQ, which is then recycled back to CoQH_2_ by FSP1 using NADPH. Indeed, NADPH was identified as a predictive biomarker of ferroptosis sensitivity in cancer cell lines [137], and its depletion by the phosphatase guanosine-3′,5′-bis(diphosphate) 3′-pyrophosphohydrolase (MESH1) facilitates ferroptosis [138]. Using Kelch-like ECH-associated protein 1 (KEAP1)–mutant lung cancer models, Koppula et al. demonstrated that the FSP1-CoQ axis is regulated by the KEAP1–nuclear factor erythroid 2–related factor 2 (NRF2) pathway [139]. The mouse double minute 2 (MDM2)–murine double minute X (MDMX) complex also regulates FSP1 through peroxisome proliferator-activated receptor α (PPARα) in a p53-independent manner to promote ferroptosis [140]. Recently, Mishima et al. discovered that vitamin K is another substrate of FSP1 that protects cells from ferroptosis [141], and they demonstrated that the reduced form of vitamin K is a potent radical-trapping antioxidant that inhibits lipid peroxidation.

### 3.3. The Dihydroorate Dehydrogenase/Glycerol-3-Phosphate Dehydrogenase 2–Mitochondrial CoQ Axis

Two recent studies uncovered mitochondria-specific ferroptosis defense mechanisms that are independent of cytosolic anti-ferroptosis pathways. In one study, Mao et al. found that dihydroorate dehydrogenase (DHODH), an enzyme required for pyrimidine synthesis, reduces CoQ to CoQH_2_ in the mitochondrial inner membrane to protect cells from ferroptosis [132]. The same group also found that glycerol-3-phosphate dehydrogenase 2 (GPD2) is another enzyme that couples glycerol-3-phosphate oxidation with mitochondrial CoQ reduction to suppress ferroptosis [133]. Importantly, these mitochondria-specific machineries cooperate with mitochondria-localized long GPX4, but not short GPX4, to suppress mitochondrial membrane lipid peroxidation to protect cells from GPX4 inhibition–induced ferroptosis. These studies suggest the importance of subcellular compartmentalization in ferroptosis defense mechanisms [142].

### 3.4. The GTP Cyclohydrolase 1–Tetrahydrobiopterin–Dihydrofolate Reductase Axis

GTP cyclohydrolase 1 (GCH1) is another GPX4-independent regulator of ferroptosis [20,143]. It protects cells from lipid peroxidation through the production of tetrahydrobiopterin (BH_4_) and dihydrobiopterin (BH_2_). BH_4_ and BH_2_ are endogenous radical-trapping antioxidants and are generated by dihydrofolate reductase (DHFR). BH_4_ is also involved in the synthesis of reduced CoQH_2_ and the depletion of PUFA-containing phospholipids.

## 4. Regulation of Ferroptosis by p53

The “guardian of the genome”, p53, is a well-established tumor suppressor. Owing to deletion and/or loss-of-function mutations, p53 has deficient function in approximately half of human cancers [144]. Many other cancers overexpress MDM2 or MDMX, which results in degradation of p53 [145]. Patients who have AML with TP53 mutations—an AML subtype that is one of the most challenging to treat—have low treatment response rates and poor survival [146]. In addition to its canonical roles of inducing cell cycle arrest, senescence, apoptosis, and DNA damage repair, p53 regulates multiple nodes of cellular metabolism, including iron, lipid, and redox metabolism, and thus plays important roles in the regulation of ferroptosis.

In 2015, Jian et al. reported that functional p53 directly suppresses SLC7A11 to induce ferroptosis and inhibit tumor development [147]. They demonstrated that acetylation-deficient TP53^3KR^ mutants can efficiently induce ferroptosis while losing their ability to upregulate canonical p53 target genes (e.g., CDKN1A/p21, PUMA). Another study showed that the mutation of all four acetylation sites in TP53 (i.e., TP53^4KR^) abolishes the gene’s effect against SLC7A11 and thus eliminates its tumor-suppressive function. These studies indicate that p53, through its regulation of senescence, apoptosis, and ferroptosis, has distinct functions as a tumor suppressor. Another study also demonstrates that mutant p53 (e.g., TP53^R175H^) retains the ability to repress the expression of SLC7A11 by inhibiting NRF2-mediated transcription programs [148]. Several metabolism-associated genes, including ALOX12 [89], the spermidine/spermine N^1^-acetyltransferase 1 gene SAT1 [149], and the ferredoxin reductase gene FDXR [150], are important for the regulation of p53-induced ferroptosis. Another p53 target, GLS2, promotes ferroptosis through the regulation of glutaminolysis; however, further investigation is needed to determine the specific role of GLS2 in p53-mediated ferroptosis [62,151].

Intriguingly, p53 can also suppress ferroptosis under certain conditions. Tarangelo et al. reported that, in contrast to the aforementioned SLC7A11 suppression and ferroptosis induction mediated by p53, the stabilization of wild-type p53 by MDM2-p53 interaction, by inducing p21 and de novo GSH synthesis, delays ferroptosis induced by system x_c_^−^ inhibition or cysteine deprivation [152]. Another direct target of p53 is calcium-independent phospholipase A2 group VI (iPLA2β), which detoxifies membrane lipid peroxidation to protect cells from ferroptosis [153]. Kuganesan et al., dissecting the roles of p53 and downstream p21, CDKs, RB, and E2F in ferroptosis modulation [154], speculated that multiple pro-ferroptosis and anti-ferroptosis signals may emanate from the individual nodes of p53 pathways, leading to context-dependent outcomes against ferroptosis stimuli and tumor progression.

Taken together, the role of p53 in ferroptosis regulation is highly complex and context dependent. Extensive reviews on this topic are available elsewhere [155,156,157].

## 5. Ferroptosis in Leukemia

### 5.1. Dysregulation of Iron Homeostasis in Leukemia

Leukemia patients are characterized by systemic iron overload due to multiple factors. The most common reason is repetitive red blood cell transfusions that lead to massive iron intake over the courses of disease [158]. Ineffective erythropoiesis and a high turnover of immature erythrocytes can also lead to iron overload, especially in AML associated with myelodysplasia. Mechanistically, ineffective erythropoiesis suppresses hepcidin leading to enhanced iron uptake [159]. Hematopoietic stem cell transplantation can also disturb iron homeostasis by suppressing erythropoiesis and by erythroid cell lysis, which can affect patients’ survival [160,161]. Lopes et al. recently reported on iron redistribution in AML cells [162]. They found that AML patients have increased transferrin saturation (TSAT) and high hepcidin irrespective of transfusion history, indicating increased levels of circulating iron. Furthermore, using electron microscopy combined with energy-dispersive X-ray spectroscopy for elemental analysis and flow cytometry for LIP quantification, they found increased intracellular iron levels in AML cells. Compared to healthy bone marrow cells, AML cells show increased expression of HO-1 and ferritin light chain 1 (FTL1), indicative of intracellular iron accumulation. The expression level of the iron importer TFR1 in leukemia cells is similar to that in healthy bone marrow cells but lower than that in erythroblasts, which suggests that leukemia cells have a relatively low iron demand [162,163].

The correlation of dysregulated iron metabolism and patient outcome in leukemia, however, remain unclear. The low expression level of the iron exporter FPN1 in AML cells is associated with chemosensitivity and better patient outcome [164]. In another study, Trujillo-Alonso et al. showed that the expression levels of FPN1 in primary AML blasts and leukemia stem cells are lower than that in healthy bone marrow CD34^+^ hematopoietic stem progenitor cells [165]. They demonstrated that low FPN1 expression leads to increased intracellular iron and oxidative stress, which sensitized cells to the iron oxide nanoparticle ferumoxytol. Together, these data suggest that leukemia cells have impaired iron flux and intracellular iron accumulation, which may render them vulnerable to ferroptosis induction.

### 5.2. Ferroptosis-Related Gene Signatures in Leukemia

Since ferroptosis was first identified as a distinct type of cell death, numerous studies have identified molecular regulators of this complicated cell death process. This extensive knowledge is organized in FerrDb, a publicly available database of ferroptosis-related genes and their associated diseases [166,167]. High-throughput gene testing with microarrays and, more recently, RNA sequencing, have enabled researchers to capture gene expression patterns as gene signatures that can be used to identify multiple clusters among patients with heterogeneous cancers based on their genetic backgrounds and to predict prognosis [168].

On the basis of these patient datasets, including those in the Cancer Genome Atlas (TCGA), multiple groups have defined ferroptosis-related genes (FRGs) and have correlated their expression with patient survival to develop FRG scoring models that can predict prognosis and ex vivo drug sensitivities in association with their clinical characteristics in AML [169,170,171,172,173,174,175,176], acute lymphoblastic leukemia (ALL) [177], and CLL [178,179]. Interestingly, many FRG signatures are associated with immune-related pathways and immune cell infiltration predicted by CIBERSORTx gene-expression profiling analyses, which suggests that ferroptosis has relevance in anti-tumor immune responses and potentially in immunotherapy in leukemia. In addition to FRGs’ expression levels, their methylation status [180] and copy number variations [181] have also been reported to have prognostic significance. One group studying the correlation of long non-coding RNAs (lncRNAs) and FRGs identified ferroptosis-related lncRNA signatures that predict the prognosis of AML [182].

Although multiple FRG signatures share some genes (e.g., GPX4, AIFM2), these models are based on different datasets and statistical methods, which has resulted in distinct scoring systems with individual clinical relevance. Further validations in independent datasets are warranted to ensure the robustness of a model to be applied in clinical practice, including in terms of its prognosis prediction and treatment optimization. In addition, basic in vitro and in vivo experiments are needed to verify the expression level and determine the biological role(s) of each FRG. Of note, ferroptosis-related prognostic gene signatures developed from the pediatric AML dataset in the National Cancer Institute’s Therapeutically Applicable Research to Generate Effective Treatments (TARGET) Initiative could not be validated in an adult AML dataset from TCGA [183], likely because of the two cohorts’ distinct genetic landscapes and clinical characteristics.

### 5.3. Ferroptosis Induction as a Therapeutic Strategy in Leukemia

Ferroptosis induction has been demonstrated to exert anti-tumor effects in various leukemia models (Table 2). Most of the evidence is based on in vitro experiments using cell lines or patient-derived primary cells, but some of the studies include in vivo models. Not only classic ferroptosis inducers (e.g., system x_c_^−^/GSH/GPX4 inhibitors) but also natural compound derivatives and other small molecules are shown to induce ferroptosis as a mechanism of their anti-leukemia activities.

#### 5.3.1. Inhibition of System x_c_^−^—GSH—GPX4 Axis in Leukemia

The first study of ferroptosis in AML was reported in 2015 by Yu et al., who demonstrated that the system x_c_^−^ inhibitor erastin induces ferroptosis in HL-60 cells in vitro [184]. The cell death was a mixture of ferroptosis and necroptosis, as it was blocked not only by ferrostatin-1 and deferoxamine but also by necrostatin-1 and the knockdown of receptor-interacting protein 3 (RIP3). They also demonstrated the association of autophagy and p38 signaling, the inhibition of which attenuated the anti-leukemia effects of erastin. Later, erastin-induced ferroptosis in HL-60 cells was shown to depend on the cytoplasmic translocation of high mobility group box 1 (HMGB1) from the nucleus, the knockdown of which attenuated ferroptosis in vivo [185]. Recently, Pardieu et al. reported that the xCT gene SLC7A11 is a putative therapeutic vulnerability, especially in NPM1-mutated AML, and a poor prognostic factor [113]. They demonstrated that the system x_c_^−^ inhibitor sulfasalazine suppressed GSH and induced oxidative stress–dependent cell death partly through ferroptosis, as indicated by its partial rescue by ferrostatin-1. The combination of sulfasalazine with daunorubicin and cytarabine was found to synergistically exert anti-leukemia effects in a patient-derived xenograft model as well as in primary AML cells. A clinical trial of sulfasalazine in combination with intensive chemotherapy in AML patients is being initiated (NCT05580861).

Zhu et al. tested erastin-induced ferroptosis in ALL cell lines, demonstrating that the majority of T-ALL cells responded poorly to in vitro erastin treatment [186]. They revealed that autophagy-mediated upregulation of voltage-dependent anion channel 3 (VDAC3) promotes ferroptosis and that the activation of autophagy by rapamycin exerts synergistic anti-leukemia effects with erastin in vivo. Another study demonstrates that progestin and adipoQ receptor 3 (PAQR3) enhances erastin- as well as RSL3-induced ferroptosis in ALLcells through the degradation of NRF2 [206]. Recently, an extensive whole-genome CRISPR knockout screen of 7 B-ALL cell lines revealed system x_c_^−^—GSH—GPX4 axis to be a common therapeutic vulnerability in B-ALL, which is partially attributed to low levels of GSH and FSP1 in these cells [187]. Targeting system x_c_^−^—GSH—GPX4 axis by RSL3, erastin, or sulfasalazine potently induced ferroptosis in B-ALL cells in vitro. FSP1 downregulation and GSH dependency in ALL cells were also reported in another study, in association with FSP1 promoter hypermethylation [207].

K562 chronic myeloid leukemia cells were tested for cysteine depletion-induced ferroptosis [188]. The authors demonstrated that the concomitant inhibition of thioredoxin reductase 1 (TXNRD1) by auranofin leads to ferroptosis in these cells. Interestingly, K562 cells with imatinib resistance (K562/G0 cells) showed higher sensitivity to ferroptosis induced by cysteine depletion.

#### 5.3.2. Ferroptosis Induction by Clinically Available Anti-Leukemia Agents

Hypomethylating agents are used as a standard of care for elderly patients with AML and for patients with myelodysplastic syndromes (MDS) [208]. One hypomethylating agent, decitabine, has been shown to induce ferroptosis and necroptosis in MDS-derived primary cells as well as cell lines [189]. It downregulates GSH and GPX4 while inducing ROS and cell death, which can be blocked by ferrostatin-1, necrostatin-1, and z-VAD-FMK. These data indicate that ferroptosis is at least partially involved in the effect of decitabine.

Wang et al. investigated the anti-leukemia effects observed for the combination of granulocyte-colony stimulating factor (G-CSF) and thrombopoietin (TPO) with low-dose chemotherapy investigated in a phase 2 trial in elderly AML patients [209]. They demonstrate that TPO induces ferroptosis through the suppression of E1A binding protein P300 (EP300)–mediated GPX4 transcription, whereas G-CSF induces pyroptosis through neutrophil elastase, which activates gasdermin D (GSDMD) in AML cells [190].

In 2020, Birsen et al. reported that, independent of its postulated effects against mutant TP53, APR-246 induces ferroptosis in AML cells during the early phases of drug exposure [191]. Ferroptosis induction by APR-246 was later confirmed in esophageal cancer cell lines, which showed an increase in GSH turnover and the suppression of mitochondrial iron-sulfur cluster biosynthesis through NFS1 [210].

Imetelstat is a first-in-class telomerase inhibitor currently under a phase 2 clinical trial against AML (NCT05583552), in addition to phase 3 clinical trial against myelodysplastic syndromes and myelofibrosis. A functional genetic screen combined with lipidomics revealed that imetelstat promotes PUFA-containing phospholipid synthesis in ACSL4– and FADS2–dependent manner, leading to AML cell ferroptosis both in vitro and in vivo [192]. However, it has not yet been shown whether the induced ferroptosis is telomerase-dependent or drug-specific off-target effects.

The tyrosine kinase inhibitor Neratinib was approved by the U.S. Food and Drug Administration in 2017 for the treatment of breast cancer, and a phase 1/2 clinical trial of this drug in pediatric patients with relapsed or refractory cancer including leukemia is ongoing (NCT02932280). Recently, neratinib has been shown to induce autophagy-dependent ferroptosis as well as G0/G1 arrest and apoptosis in HL-60 cells [193].

#### 5.3.3. Ferroptosis Induction by Natural Compounds and Their Derivatives

Many natural compounds have been studied for their ability to induce ferroptosis in cancer cells [211]. Dihydroartemisinin is a derivative of *Artemisia annua*, a plant native to China, and has anti-tumor effects in many types of cancers. This compound has effects against HL-60 cells in vitro and in vivo through the inhibition of mitochondrial oxidative phosphorylation and the activation of AMPK to induce ferritinophagy and ferroptosis [194]. Typhaneoside, a flavonoid extracted from *Pollen typhae*, has also been reported to induce apoptosis and ferroptosis associated with ferritinophagy in AML cells [195]. Hydnocarpin D is another flavonoid demonstrated to induce apoptosis and ferroptosis through autophagy in ALL cells [196]. Perillaldehyde, the main component of *Ammodaucus leucotrichus*, downregulates GSH and GPX4 to induce ferroptosis in HL-60 cells and primary AML cells [197]. Glycyrrhetinic acid is a bioactive compound of licorice, and its nanoparticles have been demonstrated to induce ferroptosis in AML cells through GPX4 downregulation [198]. These nanoparticles have synergistic anti-leukemia effects with ferumoxytol, an iron oxide nanoparticle, and programmed cell death ligand 1 (PD-L1) antibody treatment. The isothiocyanate sulforaphane (SFN) is derived from cruciferous vegetables and is known to exert anti-cancer activities. Greco et al. reported that low-dose SFN induces apoptosis whereas high-dose SFN induces ferroptosis in AML cell lines [199]. Interestingly, SFN activates the apoptosis pathway when ferroptosis is impaired, indicating that these two cell death modes could be convertible. Honokiol, a derivative of the magnolia tree, induces cell death in AML cell lines; this cell death was triggered by the upregulation of HO-1 and associated with lipid peroxidation and the alteration of ferroptosis pathway genes [200]. 4-Amino-2-trifluoromethyl-phenyl retinate (ATPR), a retinoid derivative synthesized from all-trans retinoic acid (ATRA), has been shown to activate NCOA4-dependent ferritinophagy and induce ferroptosis in AML in vitro and in vivo [201]. Poricoic acid A, the component of the mushroom *Poria cocos*, exerts anti-ALL effects in vitro and in vivo through mitochondrial dysfunction and activation of AMPK-mTOR autophagy pathway, leading to apoptosis and ferroptosis in T-ALL cells [202].

#### 5.3.4. Other Potential Therapeutic Strategies Inducing Ferroptosis in Leukemia

ALDH3a2, an enzyme that oxidizes long-chain aliphatic aldehydes, which are byproducts of lipid peroxidation, protects AML cells from oxidative stress. The inhibition of ALDH3a2 induces ferroptosis and exerts synergistic anti-leukemia effects with GPX4 inhibition or standard chemotherapy with cytarabine plus daunorubicin in vivo [96]. In addition, ALDH3a2 is selectively essential in leukemia progenitor cells but not in their normal counterparts, possibly owing to increased oxidative stress in leukemia cells.

Gold nanoparticles have anti-leukemia effects in part through ferroptosis induction. For example, GNR-CSP12 (gold nanorods loaded with chitosan and a 12-mer peptide) can induce ferroptosis through the suppression of global m^6^A RNA methylation and its combinatorial effects with tyrosine kinase inhibitors or PD-L1 checkpoint inhibitors [203].

Non-coding RNAs have also been reported to be involved in ferroptosis regulation in AML cells. A nuclear long non-coding RNA (lncRNA), LINC00618, is downregulated in AML, and its induction by vincristine treatment activates ferroptosis and apoptosis through SLC7A11 downregulation and BAX upregulation as well as caspase-3 cleavage [204]. The circular RNA circKDM4C, which is also downregulated in AML, sequesters the microRNA hsa-let-7b-5p and upregulates its downstream target, p53 [205]. The induction of circKDM4C causes AML cells to undergo ferroptosis, possibly through the downregulation of GPX4 and FTH1.

## 6. Conclusions and Future Perspectives

Recent discoveries of various non-apoptotic RCD modes, including ferroptosis, have expanded the potential modalities to induce death in cancer cells, especially in cancers resistant to conventional therapies targeting apoptosis mechanisms. A comprehensive understanding of the complex regulatory mechanisms of RCD and its involvement in cancer pathophysiology is necessary to develop RCD modes different from apoptosis into advanced cancer therapies.

While more than 10 years of extensive research has provided us with a large body of knowledge about ferroptosis, one needs to be cautious as most of our current mechanistic insights are based on in vitro models, whose environments differ from those in vivo. Components of the in vivo environment that likely affect ferroptosis regulation include oxygen; trace metals, including iron and selenium; and various metabolites such as amino acids and fatty acids. In addition, interactions with the tumor immune microenvironment (i.e., macrophages and immune cells) and even between cancer cells have been shown to affect cell vulnerability to ferroptosis, a complexity that is difficult to recapitulate in culture models. In fact, mesenchymal stem cells (MSCs) protect leukemia cells from oxidative stress through upregulation of GSH [212,213], while its significance in ferroptosis protection remains to be elucidated. Therefore, the in vitro findings must be validated in vivo to obtain a precise understanding of the mechanisms regulating ferroptosis, for which further development of specific and stable in vivo drugs to induce ferroptosis is of urgent need. The lack of definitive markers of ferroptosis is an obstacle in studying ferroptosis in vivo and in clinical settings in humans. Hence, direct evidence of ferroptosis in cancer patients treated with anti-cancer agents including ferroptosis inducers is lacking so far [214].

Since leukemia is characterized by increased oxidative stress and iron overload, one can speculate that leukemia cells are vulnerable to ferroptosis, suggesting a therapeutic potential. However, studies focusing on ferroptosis as a therapeutic modality for leukemia are limited, as discussed here. Given the physiological roles of ferroptosis, the therapeutic windows of ferroptosis induction should be carefully determined. A recent study demonstrated the potential vulnerability of hematopoietic stem cells to ferroptosis [215], warranting precise monitoring of normal hematopoiesis upon anti-cancer treatments involving ferroptosis. Taken together, implementing the concept of ferroptosis into leukemia therapeutics requires fine-tuning ferroptosis induction in normal versus malignant cells, which should be tested and established through in vivo studies and ultimately clinical trials.

In addition to understanding the underlying molecular mechanisms of ferroptosis, some of which may be specific to leukemia cells, identifying potential biomarkers that predict sensitivity or resistance to ferroptotic stimuli is critical for the development of this unique cell death mode as anti-leukemia therapy. Advancements in single-cell analysis technologies will help in the identification of potential vulnerabilities of leukemia stem/progenitor cells with specific genetic or molecular phenotypes that can be targeted for ferroptosis induction. Pursuing synthetic lethal interactions would be one promising strategy to efficiently target those populations while minimizing potential toxicity to normal cells [216]. Lastly, recent studies have revealed the interconnectivity and plasticity of different RCD pathways including ferroptosis that could compensate each other upon cell death stimuli (detailed review elsewhere [217]). Understanding the interactions as well as the molecular mechanisms of individual RCD pathways would facilitate the development of combination therapies to overcome resistance to various RCD mechanisms.

## Figures and Tables

**Figure 1 cells-12-01128-f001:**
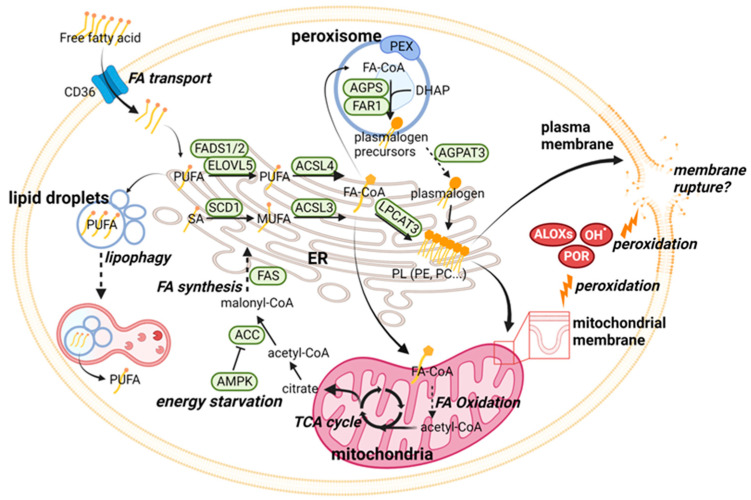
Phospholipid metabolism and its association with ferroptosis. Fatty acid biosynthesis as well as its degradation via fatty acid oxidation or lipophagy regulates ferroptosis through modulation of phospholipids. Membrane phospholipids containing polyunsaturated fatty acids are targeted by various oxidants for peroxidation leading to ferroptosis. Abbreviations—FA, fatty acid; SA, saturated fatty acid; MUFA, mono-unsaturated fatty acid; PUFA, polyunsaturated fatty acid; PL, phospholipid; ER; endoplasmic reticulum; FA-CoA, fatty acyl-CoA, DHAP, dihydroxyacetone phosphate. The image was created with BioRender (Toronto, ON, Canada) (BioRender.com).

**Figure 2 cells-12-01128-f002:**
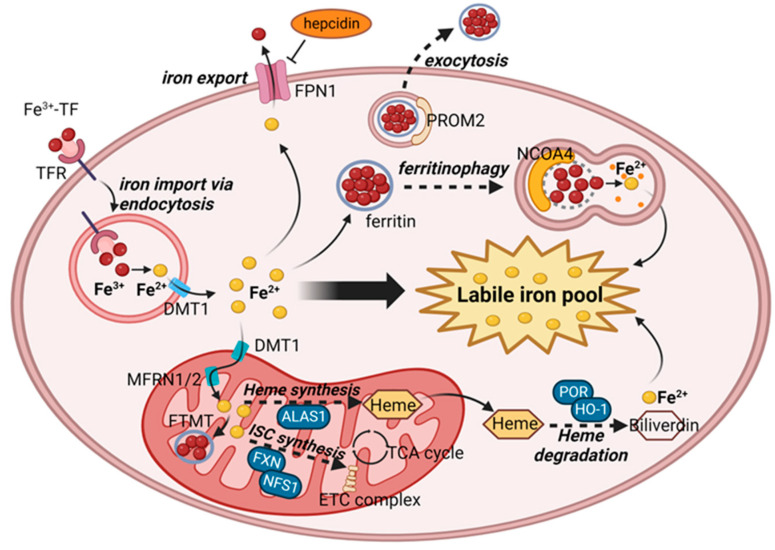
Cellular sources of labile iron pool. The level of a labile iron pool is regulated by iron import and export, storage by ferritin and its degradation (ferritinophagy), and synthesis and degradation of iron-containing proteins (heme and iron-sulfur cluster proteins). Abbreviations—ISC, iron-sulfur cluster; ETC, electron transport chain. The image was created with BioRender (BioRender.com).

**Table 1 cells-12-01128-t001:** Major players in ferroptosis regulation. Major ferroptosis-regulating molecules in each pathway are listed, with those that promote ferroptosis in bold and those that inhibit ferroptosis in yellow highlights. Abbreviations—PUFA, polyunsaturated fatty acid; MUFA, mono-unsaturated fatty acid.

Category	Pathway	Major Molecular Players
Phospholipid (PL) metabolism	Lipid peroxidation	**ALOXs, POR, PEBP1**
	PUFA-PL synthesis	**FADS1, FADS2, ELOVL5, ACSL4, LPCAT3**
	MUFA-PL synthesis	SCD1, ACSL3
	Plasmalogen synthesis	**PEX3, PEX10, AGPS, FAR1, AGPAT3**
	Energy metabolism	AMPK, **ACC**
Iron metabolism	Iron import/export	**TF, TFR1, DMT1 (import)**/FPN1 (export)
	Ferritin synthesis/export	FTH1 (synthesis)/PROM2 (export)
	Ferritinophagy	**AMPK, NCOA4**
	Mitochondrial iron storage	MFRN1, MFRN2, FTMT
	Heme synthesis/degradation	ALAS1 (synthesis)/**POR, HO-1 (degradation)**
	Iron-sulfur cluster synthesis	FXN, NFS1
Cell-cell contact	Hippo pathway	CDH1, YAP, TAZ
Ferroptosis protection mechanisms (Section 3)	System x_c_^−^—GSH—GPX4 axis	SLC7A11, GPX4
	GSH synthesis	γGCS, GSS
	Cyst(e)ine metabolism	CBS, SAHH
	Glutamine metabolism	**SLC1A5, GLS2, GOT1**
	mTORC pathway	mTORC1
	FSP1-CoQ axis	FSP1, **MESH1, PPARα, MDM2, MDMX**
	NRF2 pathway	NRF2, KEAP1
	DHODH/GPD2-CoQ axis	DHODH, GPD2
	GSH1-BH4-DHFR axis	BH2, BH4, DHFR

**Table 2 cells-12-01128-t002:** Ferroptosis-targeting studies in leukemia. Disease models (cell lines, primary cells, and/or PDX cells), experimental settings for each cell model (in vitro and/or in vivo), ferroptosis inducer(s) studied, mechanism of cell death including ferroptosis and other cell death modes, and potential combinatorial strategies are presented. Abbreviations—PDX, patient-derived xenograft; TXNRD1, thioredoxin reductase 1; TKI, tyrosine kinase inhibitor; ETC, electron transport chain.

	Models		In Vitro	In Vivo	Ferroptosis Inducers	Molecular Mechanisms of Cell Death Including Ferroptosis and Potential Combination Strategies	Ref.
System x_c_^−^/GSH/GPX4 inhibitors	AML	HL-60	●		Erastin	Mixed modes of cell death with necroptosis JNK and p38 activation Combination with AraC + daunorubicin	[184]
	AML	HL-60	●	●	Erastin	Mixed modes of cell death with apoptosis JNK and p38 activation Nuclear translocation of HMGB1	[185]
	AML	Cell lines Primary cells PDX cells	● ●	●	Sulfasalazine	Oxidative stress–induced cell death including ferroptosis Combination with AraC + daunorubicin	[113]
	ALL	Cell lines	●	●	Erastin, IKE	Combination with rapamycin-induced autophagy	[186]
	B-ALL	Cell lines PDX cells	● ●		Erastin, sulfasalazine, RSL3, FIN56	High sensitivity to ferroptosis due to low FSP1 expression	[187]
	CML	K562	●		Cysteine depletion	Combination with auranofin-induced TXNRD1 inhibition	[188]
Clinically available agents	MDS	Cell lines Primary cells	● ●		Decitabine	GSH/GPX4 inhibition–induced ferroptosis + necroptosis Combination with system x_c_^−^ inhibition (erastin)	[189]
	AML	Cell lines	●	●	Thrombopoietin	Suppression of GPX4 transcription through EP300 inhibition Combination with G-CSF-induced pyroptosis	[190]
	AML	Cell lines Primary cells	● ●	●	APR-246	GSH inhibition Combination with system x_c_^−^ inhibition or GPX4 inhibition	[191]
	AML	Cell lines PDX cells	●	● ●	Imetelstat	Enhanced PUFA-phospholipid synthesis Combination with AraC + daunorubicin	[192]
	AML	HL-60	●		Neratinib	Mixed modes of cell death with apoptosis FTH1 downregulation and cellular iron accumulation Autophagy	[193]
Natural compounds and their derivatives	AML	Cell lines Primary cells	● ●	●	Dihydroartemisinin	Mixed modes of cell death with apoptosis AMPK-mediated ferritinophagy and increased cellular iron Mitochondrial oxidative stress and ETC inhibition	[194]
	AML	Cell lines Primary cells	● ●	●	Typhaneoside	Mixed modes of cell death with apoptosis AMPK-mediated ferritinophagy and increased cellular iron Mitochondrial oxidative stress and ETC inhibition	[195]
	T-ALL	Cell lines	●		Hydnocarpin D	Mixed modes of cell death with apoptosis Autophagy	[196]
	AML	HL-60 Primary cells	● ●		Perillaldehyde	GSH/GPX4 inhibition	[197]
	AML	Cell lines	●	●	Glycyrrhetinic acid nanoparticle	GSH/GPX4 inhibition Combination with ferumoxytol or anti-PDL1 antibody	[198]
	AML	Cell lines	●		Sulforaphane	GSH/GPX4 inhibition	[199]
	AML	Cell lines	●		Honokiol	Mixed modes of cell death with apoptosis HO-1 upregulation	[200]
	AML	Cell lines	●	●	4-Amino-2-trifluoromethyl- phenyl retinate	GSH/GPX4 inhibition NCOA4-mediated ferritinophagy NRF2 downregulation	[201]
	T-ALL	Cell lines Primary cells	● ●	●	Poricoic acid A	GSH/GPX4 inhibition–induced ferroptosis + apoptosis AMPK-mediated autophagy Mitochondrial oxidative stress and ETC inhibition	[202]
Others	AML	Cell lines Primary cells MLL/AF9 mouse AML	● ●	●	ALDH3a2 inhibition	Alteration of lipid biosynthesis and cellular composition Combination with GPX4 inhibition Combination with AraC + daunorubicin	[96]
	AML	Cell lines	●	●	GNR-CSP12	GSH/GPX4 inhibition m6A hypomethylation and suppression of downstream genes Combination with TKIs including nilotinib or anti-PD1 antibody	[203]
	AML	Cell lines	●		LINC00618	System x_c_^−^ inhibition–induced ferroptosis + apoptosis Combination with vincristine	[204]
	AML	Cell lines	●	●	circKDM4C	GPX4 inhibition p53 upregulation Combination with system x_c_^−^ inhibition (erastin)	[205]

## Data Availability

Not applicable.

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
