# Peer review of "Molecular Mechanisms of Ferroptosis and Updates of Ferroptosis Studies in Cancers and Leukemia"

_cells, 2023, doi:10.3390/cells12081128_

Round 1

Reviewer 1 Report

The review of Akiyama and colleagues extensively covers the mechanisms of ferroptosis induction and inhibition. The text is well-organized and clearly written. The discussion of recent studies related to ferroptosis in leukemia adds novelty to the work. Overall, the quality of the manuscript is good; the reviewer suggests nevertheless the following minor improvements:

1.     As the knowledge on the subject is rapidly evolving, with the addition of new players to the described mechanisms, it would be useful to provide a summarizing table, showing each molecular player and briefly stating its implication in the mechanism of ferroptosis induction/inhibition. Authors may also find a way to better connect the figures to the text to allow the reader to digest the subject more easily.

2.     A brief description of ferroptosis's physiological roles and occurrences should be added.

3.     In Table 1, it would be useful to guide the reader through the table, providing a more detailed explanation in the caption below it. Please note that In vivo is repeated twice at the top of the table. System Xc-/GPXç inhibition is indicated both at the left of the table and in the corresponding boxes: please delete it in one instance, to avoid redundancy.

4.     The authors should cite Figures and Tables within the text in appropriate locations, rather than at the end of the sub-chapter titles. Please also check punctuation and correct where necessary (e.g.: (Table 1.) (Figure.1)).

Author Response

Thank you very much for your constructive suggestions. Please see below point-by point responses.

  1. We added a table to present the major pathways and molecules involved in ferroptosis regulation as Table 1. A sentence that describes the role of mTORC1 in the table was added in lines 299-300 on page 8, with the corresponding citation. Citations for the other table and figures were also changed, as described in the below response #4.
  2. We added a sentence briefly describes the physiological roles of ferroptosis with additional citations in lines 41-43 on page 1.
  3. The caption for the table (Table 2 in the revised manuscript) was modified as in lines 515-520 on page 11. With regard to the table, we changed “in vitro” and “in vivo” into italic. “SystemXc-/GSH/GPX4 inhibition” was deleted from the right column to avoid redundancy. Other cell death mechanisms that were originally in the middle column were moved to the right column “Molecular mechanisms of cell death including ferroptosis and potential combination strategies” to make the whole table more comprehensive.
  4. All the tables and figures are now cited within the text. Sentences that briefly describe each table/figure are added or moved from other paragraphs and modified as needed.

Reviewer 2 Report

Dear Authors,

Many thanks for your nice and comprehensive review on ferroptosis and your emphasis on leukaemia. 

Expect a few typos that need to be corrected (see below), your review is worth publishing for me. Nonetheless, I would like to have your opinion on the two additional pieces of information I would like to read in your paper:

1) Are there any evidence of Ferroptosis in patients treated with chemotherapeutic drugs or Ferroptosis inducers ?

2) I would like to know whether you could add any comment related to the cancer selectivity of these inducers or their potential toxicity, given that these inducers may also allow the triggering of ferroptosis in health cells. 

Typos or sentence clarification

1) page 2 chapter 2.1, Second sentence "... PUFAs to abstract a hydrogen..." Do you mean extract ?

2) Page 9 last sentence : "... therefore it is intriguing....". I'm not sure to understand what you mean to say in this sentence. You first mention that CD44 has been found as a putative target in AML, you also mention that this type I glycoprotein is a putative iron transporter ! So why is it intriguing to try to investigate its role in iron uptake in AML ? You may need to rephrase this sentence or elaborate your thoughts ! 

3) Table 1 : change cricKDM4C to circKDM4C

Author Response

Thank you for your constructive suggestions and corrections. Please see below point-by-point responses to your comments.

1) It is a very important point but to the best of our knowledge, there is no solid evidence of ferroptosis induction in cancer patients treated with anti-cancer drugs including ferroptosis inducers. This is at least partly because of the lack of definitive markers or assays to detect ferroptosis, especially in vivo. We added some sentences describing this issue with an additional citation that discusses the translation of ferroptosis in lines 633-635 on page 15.

2) We agree that it is a critical point to be discussed, tested and addressed in this field. Selectivity of therapeutic ferroptosis induction needs to be carefully determined through in vivo experiments and clinical trials for each therapeutic modality. We modified the paragraph discussing the therapeutic window of ferroptosis induction in a little more details, in lines 639-646 on page 15.

Typos or sentence clarification

1) We changed the word “abstract” to “remove” to make it clear (line 84 on page3). We also made the same change on line 183, page5.

2) We agree that this sentence does not make sense. Although CD44 is a putative target in AML and is known as an iron transporter, there is no clear mechanistic link between therapeutic CD44 inhibition and ferroptosis induction. We removed the sentence to avoid unclarity.

3) Corrected.